# Integral Pruning on Activations and Weights for Efficient Neural Networks

## Abstract

With the rapidly scaling up of deep neural networks (DNNs), extensive research studies on network model compression such as weight pruning have been performed for efficient deployment. This work aims to advance the compression beyond the weights to the activations of DNNs. We propose the Integral Pruning (IP) technique which integrates the activation pruning with the weight pruning. Through the learning on the different importance of neuron responses and connections, the generated network, namely *IPnet*, balances the sparsity between activations and weights and therefore further improves execution efficiency. The feasibility and effectiveness of IPnet are thoroughly evaluated through various network models with different activation functions and on different datasets. With $< 0.5\%$ disturbance on the testing accuracy, IPnet saves $71.1\% \sim 96.35\%$ of computation cost, compared to the original dense models with up to $5.8\times$ and $10\times$ reductions in activation and weight numbers, respectively. The source codes are available at (*omitted for blind review*).

## 1 Introduction

Deep neural networks (DNNs) have demonstrated significant advantages in many real-world applications, such as image classification, object detection and speech recognition (He et al., 2016; Redmon et al., 2016; Sainath & Parada, 2015). On the one hand, DNNs are developed for improving performance in these applications, which leads to intensive demands in data storage, communication and processing. On the other hand, the ubiquitous intelligence promotes the deployment of DNNs in light-weight embedded systems that are equipped with only limited memory and computation resource. To reduce the model size while ensuring the performance quality, DNN pruning is widely explored. Redundant weight parameters are removed by zeroing-out those in small values (Han et al., 2015; Park et al., 2016). Utilizing the zero-skipping technique (Han et al., 2016) on sparse weight parameters can further save the computation cost. In addition, many specific DNN accelerator designs (Albericio et al., 2016; Reagen et al., 2016) leveraged the intrinsic zero-activation pattern of the rectified linear unit (ReLU) to realize the activation sparsity. The approach, however, cannot be directly extended to other activation functions, e.g., leaky ReLU.

Although these techniques achieved tremendous success, pruning only the weights or activations cannot lead to the best inference speed, which is a crucial metric in DNN deployment, for the following reasons. First, the existing weight pruning methods mainly focus on the model size reduction. However, the most essential challenge of speeding up DNNs is to minimize the computation cost, such as the intensive multiple-and-accumulate operations (MACs). Particularly, the convolution (*conv*) layers account for most of the computation cost and dominate the inference time in DNNs (Park et al., 2016). Because weights are shared in convolution, the execution speed of conv layers is usually bounded by computation instead of memory accesses (Jouppi et al., 2017; Zhang et al., 2015). Second, the activation in DNNs is not strictly limited with ReLU. The intrinsic zero-activation patterns do not exist in non-ReLU activation functions, such as leaky ReLU and sigmoid. Third, the weights and activations of a network together determine the network performance. Our experiment shows that the zero-activation percentage obtained by ReLU decreases after applying the weight pruning (Han et al., 2016). Such a deterioration in activation sparsity could potentially eliminate the advantage of the aforementioned accelerator designs.

In this work, we propose the integral pruning (IP) technique to minimize the computation cost of DNNs by pruning both weights and activations. As the pruning processes for weights and activations are correlated, IP learns dynamic activation masks by attaching activation pruning to weight pruning after static weight masks are well trained. Through the learning on the different importance of neuron responses and connections, the generated network, namely *IPnet*, balances the sparsity between activations and weights and therefore further improves execution efficiency. Moreover, our method not only stretches the intrinsic activation sparsity of ReLU, but also targets as a general approach for other activation functions, such as leaky ReLU. Our experiments on various network models with different activation functions and on different datasets show substantial reduction in MACs by the proposed IPnet. Compared to the original dense models, IPnet can obtain up to $5.8\times$ activation compression rate, $10\times$ weight compression rate and eliminate $71.1\% \sim 96.35\%$ of MACs. Compared to state-of-the-art weight pruning technique (Han et al., 2015), IPnet can further reduce the computation cost $1.2\times \sim 2.7\times$.

## 2 RELATED WORKS

**Weight Pruning:** The weight pruning emerges as an effective compression technique in reducing the model size and computation cost of neural networks. A common approach of pruning the redundant weights in DNN training is to include an extra regularization term (e.g., the $\ell_1$-normalization) in the loss function (Liu et al., 2015; Park et al., 2016) to constrain the weight distribution. Then the weights below a heuristic threshold will be pruned. Afterwards, a certain number of finetuning epochs will be applied for recovering the accuracy loss due to the pruning. In practice, the direct-pruning and finetuning stages can be carried out iteratively to gradually achieve the optimal trade-off between the model compression rate and accuracy. Such a weight pruning approach demonstrated very high effectiveness, especially for fully-connected (*fc*) layers (Han et al., 2015). For conv layers, removing the redundant weights in structured forms, e.g., the filters and filter channels, has been widely investigated. For example, Wen et al. (2016) proposed to apply *group Lasso regularization* on weight groups in a variety of self-defined sizes and shapes to remove redundant groups. Molchanov et al. (2016) used the first-order Taylor series expansion of the loss function on feature maps to determine the rankings of filters and those in low ranking will be removed. The filter ranking can also be represented by the root mean square or the sum of absolute values of filter weights (Mao et al., 2017; Yu et al., 2017).

**Activation Sparsity:** The activation sparsity has been widely utilized in various DNN accelerator designs. Chen et al. (2016), Albericio et al. (2016) and Reagen et al. (2016) accelerated the DNN inference with reduced off-chip memory access and computation cost benefiting from the sparse activations originated from ReLU. A simple technique to improve activation sparsity by zeroing out small activations was also explored (Albericio et al., 2016). However, the increment of activation sparsity is still limited without accuracy loss. The biggest issue in the aforementioned works is that they heavily relied on ReLU. However, zero activations do not exist in non-ReLU activation function. To regulate and stretch the activation sparsity, many dropout-based methods are proposed. Adaptive dropout (Ba & Frey, 2013), for instance, developed a binary belief network overlaid on the original network. The neurons with larger activation magnitude incur higher probability to be activated. Although this method achieved a better regularization on DNNs, the inclusion of belief network complicated the training and had no help on inference speedup. The winners-take-all (WTA) autoencoder was built with a regularization based on activation magnitude to learn deep sparse representations from various datasets (Makhzani & Frey (2015)).

As can be seen that the model size compression is the main focus of weight pruning, while the use of activation sparsification focuses more on the intrinsic activation sparsity by ReLU or exploring the virtue of sparse activation in the DNN training for better model generalization. In contrast, our proposed IP aims for reducing the DNN computation cost and therefore accelerating the inference by integrating and optimizing both weight pruning and activation sparsification.

## 3 APPROACH

As depicted in Figure 1, the proposed IP consists of two steps by concatenating the activation pruning to the weight pruning. Both stages seek for unimportant information (weights and activations,

respectively) and mask them off. We aim to keep only the important connections and activations to minimize the computation cost. In this section, we will first explain the integration of the two steps. The technical details in model quality (e.g., accuracy) control will then be introduced. The prediction method for deriving activation masks is also proposed to speed up the inference of IPnets. At last, the appropriate settings of dropout layers and training optimizers are discussed.

### 3.1 INTEGRATION OF WEIGHT PRUNING AND ACTIVATION PRUNING

**Weight pruning.** In the weight pruning stage, weight parameters with magnitude under a threshold are masked out, and weight masks will be passed to the following finetuning process. After the model is finetuned for certain epochs to recover accuracy loss, weight masks need to be updated for the next finetuning round. There are two crucial techniques to help weight pruning. 1) The threshold used to build weight masks are determined based on the weight distribution of each layer. Because of different sensitivity for weight pruning, each layer owns a specific weight sparsity pattern. Basically, the leading several conv layers are more vulnerable to weight pruning. 2) The whole weight pruning stage needs multiple pruning-finetuning recursions to search an optimal weight sparsity. Weight masks are progressively updated to increase pruning strength.

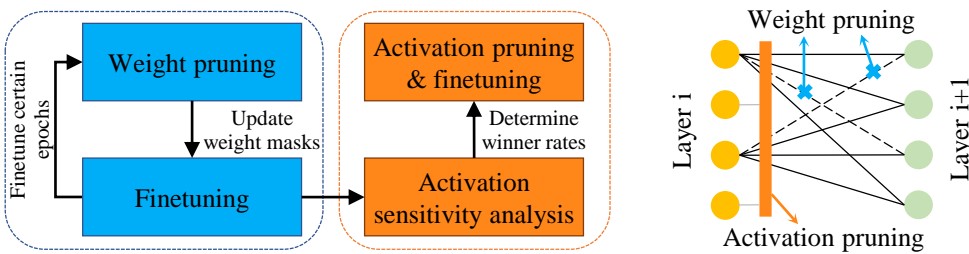

Figure 1: Working flow of integral pruning.

**Activation pruning.** While weak connections between layers are learned to be pruned, activaitons with small magnitude are taken as unimportant and can be masked out to further minimize inter-layer connections, and hence to reduce computation cost. Notice that, neurons in DNNs are trained to be activated in various patterns according to different input classes, thus **dynamic** masks should be learned in the activation pruning stage, which are different from the **static** masks in the weight pruning stage. The selected activations by the dynamic mask are denoted as *winners*, and the winner rate is defined as:

$$Winner\ rate = \frac{S_{winner}}{S_{total}},\tag{1}$$

where $S_{winner}$ and $S_{total}$ denote the number of winners and total activation number. The winner rate per layer is determined by the analysis of activation pruning sensitivity layer-wise on the models obtained after weight pruning. The winner activation after the pruning mask, $A_m$, obeys the rule:

$$A_m = \begin{cases} A_{orig}, & if\ |A_{orig}| > \theta \\ 0, & otherwise \end{cases}\tag{2}$$

where $\theta$ is the threshold derived at run-time from the activation winner rate for each layer, and $A_{orig}$ is the result from original activation function. Same with weight pruning, the model with dynamic activation masks is finetuned to recover accuracy drop. No iterative procedure of mask updating and finetuning is required in our activaiton pruning method.

### 3.2 WINNER RATE SETTINGS

Not all layers share the same winner rate. Similar to the trend in weight pruning, deeper layers tolerate larger activation pruning strength. To analyze the activation pruning sensitivity, the model with activation masks is tested on a validation set sampled from the training images with the same size as the testing set. Accuracy drops are taken as the indicator of pruning sensitivity for different winner rate settings. Before finetuning, the activation winner rate per layer is set empirically to keep accuracy drop less than 2%. For the circumstances that model accuracy is resistant to be tuned back, winner rates in the leading several layers should be set smaller. Examples of sensitivity analysis will be given and discussed in Section 5.

### 3.3 Threshold prediction in activation pruning

The dynamic activation pruning method increases the activation sparsity and maintains the model accuracy as well. The solution of determining threshold $\theta$ in Equation (2) for activation masks is actually a canonical $argpartion$ problem to find top-k arguments in an array. According to the Master Theorem (Bentley et al., 1980), argpartition can be fast solved in linear time $\mathcal{O}(N)$ through recursive algorithms, where N is the number of elements to be partitioned. To further speed up, threshold prediction can be applied on the down-sampled activation set. An alternate threshold $\theta'$ is predicted by selecting top-$\alpha k$ elements from the down-sampled activation set comprising $\alpha N$ elements with $\alpha$ as the down-sampling rate. $\theta'$ is applied for the original activation set afterwards.

### 3.4 Dropout layer with activation pruning

For DNN training, dropout layer is commonly added after large fc layers to avoid over-fitting problem. The neuron activations are randomly chosen in the feed-forward phase, and weights updates will be only applied on the neurons associated with the selected activations in the back-propagation phase. Thus, a random partition of weight parameters are updated in each training iteration. Although the activation mask only selects a small portion of activated neurons, dropout layer is still needed, for the selected neurons with winner activations are always kept and updated, which makes over-fitting prone to happen. In fc layers, the remaining activated neurons are reduced to $S_{winner}$ from $S_{total}$ neurons as defined in Equation (1). The dropout layer connected after the activation mask is suggested to be modified with the setting:

$$Dropout\ rate = 0.5\sqrt{\frac{S_{winner}}{S_{total}}} = 0.5\sqrt{Winner\ rate},\tag{3}$$

where 0.5 is the conventionally chosen dropout rate in the training process for original models, and the activation winner rate is introduced to regulate the dropout strength for balancing over-fitting and under-fitting. The dropout layers will be directly removed in the inference stage.

### 3.5 Optimizer and learning rate

We find different optimizer requirements for weight pruning and activation pruning. In the weight pruning stage, it's recommended to adopt the same optimizer used for training the original model. The learning rate should be properly reduced to $0.1\times \sim 0.01\times$ of the original learning rate. In the activation pruning stage, our experiments show that Adadelta (Zeiler, 2012) usually brings the best performance. Adadelta adapts the learning rate for each individual weight parameter. Smaller updates are performed on neurons associated with more frequently occurring activations, whereas larger updates will be applied for infrequent activated neurons. Hence, Adadelta is beneficial for sparse weight updates, which is exactly the common situation in our activation pruning. During finetuning, only a small portion of weight parameters are updated because of the combination of sparse patterns in weights and activations. The learning rate for Adadelta is also reduced $0.1\times \sim 0.01\times$ compared to that used in training the original model.

## 4 Experiments

All of our models and evaluations are implemented in TensorFlow. IPnets are verified on various models ranging from simple multi-layer perceptron (MLP) to deep convolution neural networks (CNNs) on three datasets, MNIST, CIFAR-10 and ImageNet as in Table 1. For AlexNet (Krizhevsky et al., 2012) and ResNet-32 (Zagoruyko & Komodakis, 2016), we focus on conv layers because conv layers account for more than 90% computation cost in these two models.

The compression results of IPnets on activations, weights and MACs are summarized in Table 1 compared to the original dense models. IPnets achieve a **2.3$\times$** $\sim$ **5.8$\times$** activation compression rate and a **2.5$\times$** $\sim$ **10$\times$** weight compression rate. Benefiting from sparse weights and activations, IPnets only need **3.65%** $\sim$ **28.9%** of MACs required in dense models. The accuracy drop is kept less than 0.5%, and for some cases, e.g., MLP-3 and AlexNet in Table 1, the IPnets achieve a better accuracy.

Table 1 shows that our method can learn both sparser activations and sparse weights and thus save computation. More importantly, in Figure 2, we will show that our approach is superior to ap-

proaches which explore intrinsic sparse ReLU activations and state-of-the-art weight pruning. The ReLU function brings intrinsic zero activations for MLP-3, ConvNet-5 and AlexNet in our experiments. However, the non-zero activation percentage increases in weight-pruned (WP) models as depicted in Figure 2 (a). The increment of non-zero activations undermines the effort from weight pruning. The activation pruning can remedy the activation sparsity loss and prune 7.7% - 18.5% more activations even compared to the original dense models. The largest gain from IP exits in ResNet-32 which uses leaky ReLU as activation function. Leaky ReLU generates dense activations in the original and WP models. The IPnet for ResNet-32 realizes a 61.4% activation reduction. At last, IPnets reduce $4.4\% \sim 22.7\%$ more MACs compared to WP models as depicted in Figure 2 (b), which means a **1.2× ∼ 2.7×** improvement. More details on model configuration and analysis are discussed as follows.

Table 1: Summary of IPnets

| Network | MLP-3 | ConvNet-5 | AlexNet | ResNet-32 |
|---|---|---|---|---|
| Dataset | MNIST | CIFAR-10 | ImageNet | CIFAR-10 |
| Orig Acti Function | ReLU | ReLU | ReLU | Leaky ReLU |
| Accuracy Baseline | 98.41% | 86% | 57.22% | 95.01% |
| Accuracy IP | 98.42% | 85.94% | 57.26% | 94.58% |
| Activation % | 17.1% | 43.6% | 44.2% | 38.6% |
| Weight % | 10% | 40.4% | 38.8% | 32.4% |
| MAC % | 3.65% | 27.7% | 28.9% | 13.7% |

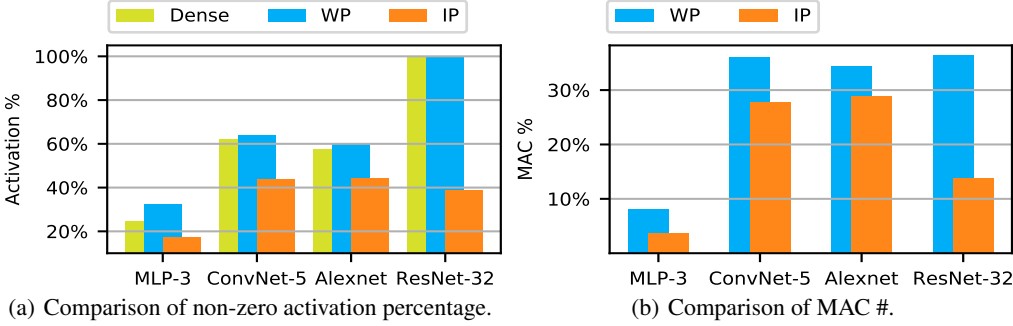

(a) Comparison of non-zero activation percentage.   (b) Comparison of MAC #.

Figure 2: Comparison between WP models and IPnets.

## 4.1 MLP-3 ON MNIST

The MLP-3 on MNIST has two hidden layers with 300 and 100 neurons respectively, and the model configuration details are summarized in Table 2. The amount of MACs is calculated with batch size as 1, and the non-zero activation percentage at the output per layer is averaged from random 1000 samples from the training dataset. The following discussions on other models obey the same statistics setting. The model size of MLP-3 is firstly compressed 10× through weight pruning. IP further reduces the total number of MACs to 3.65% by keeping only 17.1% activations. The accuracy of the priginal dense model is 98.41% on MNIST, and the aggressive reduction of MACs (27.4×) doesn't decrease the accuracy.

Table 2: MLP-3 on MNIST

| Layer | Shape | Weight # | MAC # | Acti % | Weight % | MAC % |
|---|---|---|---|---|---|---|
| fc1 | 784×300 | 235.2K | 235.2K | 12% | 10% | 3.77% |
| fc2 | 300×100 | 30K | 30K | 24% | 10% | 2.62% |
| fc3 | 100×10 | 1K | 1K | 100% | 20% | 6.81% |
| Total | | 266.2K | 266.2K | 17.1% | 10% | 3.65% |

## 4.2 CONVNET-5 ON CIFAR-10

For digit images in MNIST dataset have specific sparse features, the results on small-footprint MLP-3 are very promising. IP is further applied for a 5-layers CNN, ConvNet-5, on a more complicated dataset, CIFAR-10. With two conv layers and three fc layers, the original model has an 86% accuracy. As shown in Table 3, the IPnet for ConvNet-5 only needs 27.7% of total MACs compared to the dense model through pruning 59.6% of weights and 56.4% of activations at the same time. The accuracy only has a marginal 0.06% drop. The dominant computation cost is from conv layers accounting for more than $4/5$ of total MACs for inference. Although fc layers can generally be pruned in larger strength than conv layers, the computation cost reduction of IPnet is dominated by the pruning results in conv layers.

Table 3: ConvNet-5 on CIFAR-10

| Layer | Shape | Weight # | MAC # | Acti % | Weight % | MAC % |
|-------|-------|----------|-------|--------|----------|-------|
| conv1 | 5×5, 64 | 4.8K | 0.69M | 50.6% | 70% | 70% |
| conv2 | 5×5, 64 | 102.4K | 3.68M | 17.3% | 50% | 25.3% |
| fc1 | 2304×384 | 884.7K | 884.7K | 9.9% | 40% | 6.92% |
| fc2 | 384×192 | 73.7K | 73.7K | 44.8% | 30% | 3% |
| fc3 | 192×10 | 1.92K | 1.92K | 100% | 50% | 22.4% |
| Total | | 1.07M | 5.34M | 43.6% | 40.4% | 27.7% |

## 4.3 ALEXNET ON IMAGENET

We push IP onto AlexNet for ImageNet ILSVRC-2012 dataset which consists of about 1.2M training images and 50K validating images. The ALexNet comprises 5 conv layers and 3 fc layers and achieves 57.22% top-1 accuracy on the validation set. Similar to ConvNet-5, the computation bottle-neck of AlexNet exits in conv layers by consuming more than $9/10$ of total MACs. We focus on conv layers here. As shown in Table 4, deeper layers have larger pruning strength on weights and activations because of the sparse high-level feature abstraction of input images. For example, the MACs of layer conv5 can be reduced $10\times$, while only a $1.2\times$ reduction rate is realized in layer conv1. In total, the needed MACs are reduced $3.5\times$ using IP with 38.8% weights and 44.2% activations.

Table 4: AlexNet on ImageNet

| Layer | Shape | Weight # | MAC # | Acti % | Weight % | MAC % |
|-------|-------|----------|-------|--------|----------|-------|
| conv1 | 11×11, 96 | 34.85K | 112.2M | 68.7% | 85% | 85% |
| conv2 | 5×5, 256 | 307.2K | 240.8M | 35.8% | 40% | 27.5% |
| conv3 | 3×3, 384 | 884.7K | 149.5M | 25% | 35% | 12.6% |
| conv4 | 3×3, 384 | 663.5K | 112.1M | 25% | 40% | 10% |
| conv5 | 3×3, 256 | 442.4K | 74.8M | 27.7% | 40% | 10% |
| Total | | 2.33M | 689.5M | 44.2% | 38.8% | 28.9% |

## 4.4 GOING DEEPER

CNN models are getting deeper with tens to hundreds of conv layers. We verify the IP method on ResNet-32 as shown in Table 5. The ResNet-32 consists of 1 conv layer, 3 stacked residual units and 1 fc layer. Each residual unit contains 5 consecutive residual blocks. The filter numbers in residual units increase rapidly, and same for weight amount. An average pooling layer is connected before the last fc layer to reduce feature dimension. Compared to conv layers, the last fc layer can be neglected in terms of weight volume and computation cost. The original model has a 95.01% accuracy on CIFAR-10 dataset with 7.34G MACs per image. Weight and activation pruning strength is designed unit-wise to reduce the exploration space of hyperparameters, i.e., threshold settings. Notice that leaky ReLU is used as the activation function, thus zero activations are extremely hard to occur in the original and WP model. Only with IP, the activation percentage can be reduced down to 38.6%. As shown in Table 5, the model size is compressed $3.1\times$, and the final gain is that 86.3% of MACs can be avoided while keeping the accuracy drop less than 0.5%.

By randomly selecting 500 images from the training images, the activation distribution of the first residual block in baseline model is depicted in Figure 3 (a). Activations gather near zero with

Table 5: ResNet-32 on CIFAR-10

| Layer | Shape | | | Weight # | MAC # | Acti % | Weight % | MAC % |
|-------|-------|---|---|----------|-------|--------|----------|-------|
| conv1 | 3×3, 16 | | | 0.43K | 0.44M | 40% | 40% | 40% |
| unit2 | { | 3×3, 160
3×3, 160 | } × 5 | 2.1M | 2.15G | 40% | 40% | 16% |
| unit3 | { | 3×3, 320
3×3, 320 | } × 5 | 8.76M | 2.6G | 40% | 40% | 16% |
| unit4 | { | 3×3, 640
3×3, 640 | } × 5 | 35.02M | 2.6G | 30% | 30% | 9.5% |
| Total | | | | 45.87M | 7.34G | 38.6% | 32.4% | 13.7% |

long tails towards both positive and negative directions. The activation distribution after IP are shown in Figure 3 (b). Activations near zero are pruned out, and the major contribution comes from removing small negative values. In addition, the kept activations are trained to be stronger with larger magnitude, which is consistent with the phenomenon that the non-zero activation percentage increases after weight pruning when using ReLU as illustrated in Figure 2 (a).

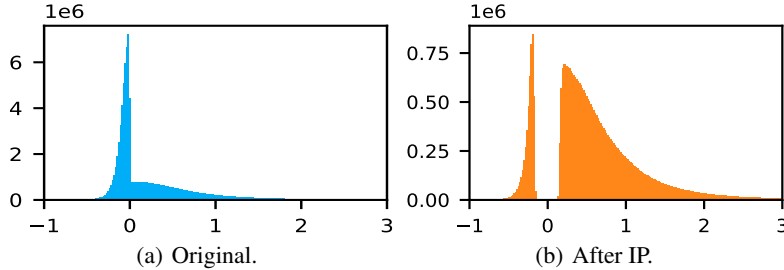

(a) Original.      (b) After IP.

Figure 3: Activation distribution of ResNet-32 (500 image samples).

## 5 DISCUSSION

The static activation pruning approach has been widely adopted in efficient DNN accelerator designs (Albericio et al., 2016; Reagen et al., 2016). By selecting a proper static threshold $\theta$ in Equation (2), more activations can be pruned with little impact on model accuracy. For the activation pruning in IP, the threshold is dynamically set according to the winner rate and activation distribution layer-wise. The comparison between static and dynamic pruning is conducted on ResNet-32 for CIFAR-10 dataset. For the static pruning setup, the $\theta$ for leaky ReLU is assigned in the range of [0.07, 0.14], which brings different activation sparsity patterns.

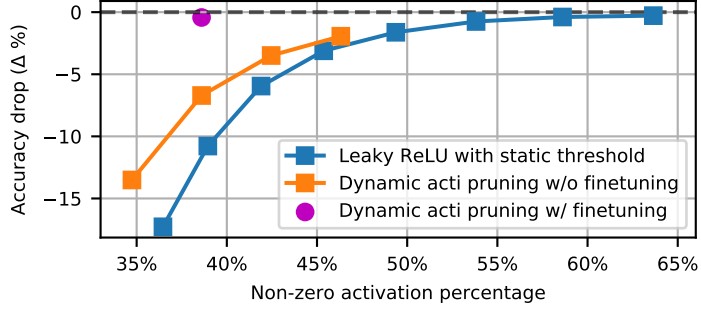

Figure 4: Comparison to static activation pruning.

As the result of leaky ReLU with static threshold shown in Figure 4, the accuracy starts to drop rapidly when non-zero activation percentage is less than 58.6% ($\theta = 0.08$). Using dynamic threshold settings according to winner rates, a better accuracy can be obtained under the same activation sparsity constraint. Finetuning the model using dynamic activation masks will dramatically recover

the accuracy loss. As our experiment in Section 4.4, the IPnet for ResNet-32 can be finetuned to eliminate the 10.4% accuracy drop caused by the static activation pruning.

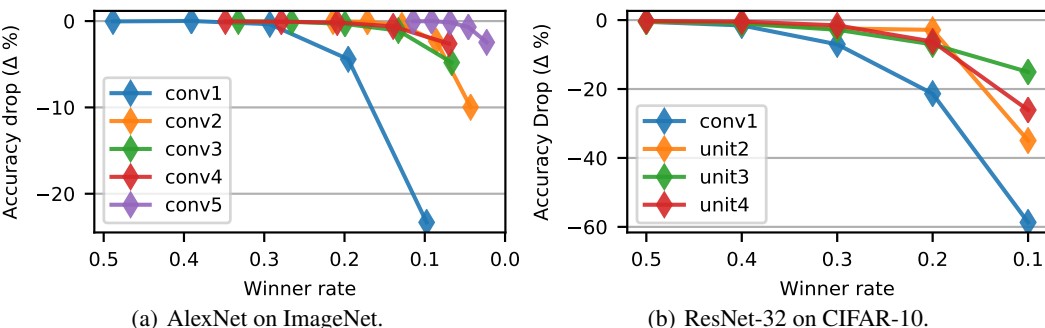

Figure 5: Activation pruning sensitivity.

In weight pruning, the applicable pruning strength is different per layer (Han et al., 2015; Molchanov et al., 2016). Similarly, the pruning sensitivity analysis is required to determine the proper activation pruning strength layer-wise, i.e., the activation winner rate per layer. Figure 5 shows two examples on WP models from AlexNet and ResNet-32. For AlexNet in Figure 5 (a), the accuracy drops sharply when the activation winner rate of layer conv1 is less than 0.3. Meanwhile, the winner rate of layer conv5 can be set under 0.1 without hurting accuracy. Deeper conv layers can support sparser activations. The ResNet-32 in Figure 5 (b) has a similar trend of activation pruning sensitivity. Layer conv1 is most susceptible to the activation pruning. Verified by thorough experiments in Section 4, the accuracy loss can be well recovered by finetuning with proper activation winner rates.

As discussed in Section 3.3, the process to select activation winners can be accelerated by threshold prediction on down-sampled activation set. We apply different down-sampling rates on the IPnet for AlexNet. As can be seen in Figure 6, layer conv1 is most vulnerable to threshold prediction. From the overall results, it's practical to down-sample 10% ($\alpha = 0.1$) of activations by keeping the accuracy drop less than 0.5%.

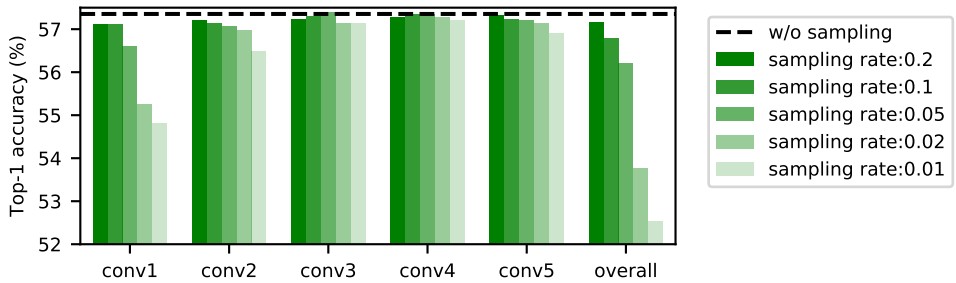

Figure 6: The effects of threshold prediction.

## 6 CONCLUSION

To minimize the computation cost in DNNs, IP combining weight pruning and activation pruning is proposed in this paper. The experiment results on various models for MNIST, CIFAR-10 and ImageNet datasets have demonstrated considerable computation cost reduction. In total, a 2.3× - 5.8× activation compression rate and a 2.5× - 10× weight compression rate are obtained. Only 3.65% - 28.9% of MACs are left with marginal effects on model accuracy, which outperforms the weight pruning by 1.2× - 2.7×. The IPnets are targeted for the dedicated DNN accelerator designs with efficient sparse matrix storage and computation units on chip. The IPnets featuring compressed model size and reduced computation cost will meet the constraints from memory space and computing resource in embedded systems.

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
