# OpenReview forum: "Integral Pruning on Activations and Weights for Efficient Neural Networks"
_ICLR.cc/2019/Conference_

### Official Review · AnonReviewer3 · 2018-11-02
**Pruning of weights and activations**

**Rating:** 5
**Confidence:** 4

**Review:**

This article presents a novel approach called Integral Pruning (IP) to reduce the computation cost of Deep neural networks (DNN) by integrating activation pruning along with weight pruning. The authors show that common techniques of exclusive weight pruning does compress the model size, but increases the number of non-zero activations after ReLU. This would counteract the advantage of DNN accelerator designs (Albericio et al., 2016; Reagen et al., 2016) that leverage activation sparsity to speed up the computations. IP starts with pruning the weights using an existing technique to mask out weights under a threshold and then fine-tune the network in an iterative fashion to maintain the accuracy. After weight pruning, IP further masks out the activations with smaller magnitude to reduce the computation cost. Unlike weight pruning techniques that use static masks, the authors propose to use dynamic activation masks for activation sparsity in order to account for various patterns that are being activated in DNN for different input samples. In order to do this, the 'winner rate' measure for every layer (or for a group of layers in deep networks like ResNet32) is defined, to dynamically set the threshold for the generation of activation masks which eventually controls the amount of non-zero activation entries. The article empirically analyzes the sensitivity of activation pruning on validation data by setting different winner rates at every layer in DNN and decides upon a set of winner rates accordingly followed by an iteration of fine-tuning the network to maintain its performance. The authors show that their technique produced lower number of non-zero activations in comparison with the intrinsic sparse ReLU activations and weight pruning techniques.

The topic of reducing network complexity for embedded implementations of DNNs is highly relevant, in particular for the ICLR community.

The IP technique yields significantly reduced number of multiply-accumulate operations (MACs) across different models like MLP-3, ConvNet-5, ResNet32 and AlexNet and on different datasets like MNIST, CIFAR10 and ImageNet. They also depicted that pruning the activations with dynamic activation masks followed by fine-tuning the network yields more sparse activations and negligible loss in accuracy when compared against using static activation masks.


Strengths of the paper:
- The motivation to extend compression beyond the weights to activations in order to support the DNN accelerator designs and the technical details are clearly explained.
- The proposed technique indeed produces sparser activations than intrinsic ReLu sparse activations and can also applied to any network regardless of the choice of activation function.
- The proposed technique is evaluated across different network architectures and datasets.
- The advantage of adapting dynamic activation masks over static ones is clearly demonstrated.

 Weaknesses of the paper:
- The originality of the approach is limited because it is a relatively straightforward combination of existing techniques for weight and activation pruning.
- The "winner rate" measure is defined for every layer and should be explored over different values in order to find the equilibrium to reduce the number of non-zero activations and maintain the accuracy. This search of winner rates will become inefficient as the depth of the network increases. However, the authors used a single winner rate for a group of layers in case of ResNet-32 to reduce the exploration of search space but this choice might lead to suboptimal results.
- The authors compare the resultant number of MAC operations against numbers from the weight pruning technique. However, there also exist different works on group pruning techniques like Liu et al. (2017), Huang & Wang (2017), Ye et al. (2018) to prune entire channels / feature maps and thus yield more compact networks. Since these approaches prune the channels, they show a direct impact on the computation complexity and greatly reduce the computation time. A proper and fair comparison would be to compare the numbers of IP against such group pruning techniques. This comparison is highly important to highlight the significance of the approach on speeding up the DNNs and it is missing from the paper.
- At several locations in Section 4, e.g. Sec. 4.1, 4.3, and 4.4. there is no precise statement about the incurred accuracy loss (or no statement at all). The connection to Figures 4 and 5 is not immediately clear and should be made explicit.

References:
- Zhuang Liu, Jianguo Li, Zhiqiang Shen, Gao Huang, Shoumeng Yan, and Changshui Zhang. Learning efficient convolutional networks through network slimming.
- Jianbo Ye, Xin Lu, Zhe Lin, and James Z Wang. Rethinking the smaller-norm-less-informative assumption in channel pruning of convolution layers
- Zehao Huang and Naiyan Wang. Data-driven sparse structure selection for deep neural networks.

Overall Evaluation:
The authors integrate activation pruning along with the weight pruning and show that the number of MAC operations are greatly reduced by their technique when compared to the numbers of weight pruning alone. 	However, I am not convinced regarding the reported number of MAC operations since the number of MAC operations of sparse weight matrices and activations would remain the same as the original models unless some of the filters/activation maps are pruned from the network.  On the other hand, comparisons against group pruning techniques are highly necessary to evaluate the potential impact of the approach on speeding up of DNNs. My preliminary rating is a weak reject but I am open to revise my rating based on the authors response to the above stated major weaknesses.

Minor comments:
- Caption of Fig. 4 should mention the task on which the results were obtained.
- There are occasional grammar errors and typos that should be corrected.

---

> ### Author Response · Authors · 2018-11-12
> **Novelty, Winner Rate Layer-wise, and Comparison**
>
> Thanks a lot for your suggestions.
>
> -- The originality of the approach
>
> While weight pruning technique adopted in this paper is on-the-shelf, the dynamic activation pruning is first proposed here. Before our work, the research focus is on the static structured/unstructured weight pruning.
>
> -- Winner Rate Layer-wise
>
> Thanks for your deep insight here. We agree with your concern about potential suboptimal results. To be honest, the direct motivation to do winner rates searching was to make the table concise to fit page limit. We will add more experiments here and in the Appendix.
>
> First, we answer the question about searching complexity of winner rates. It takes about 20min to complete the winner rate scanning for ResNet-32 group-wise as shown in Fig.5 (b). For exploring winner rates layer-wise, it takes us about 2 hours. When setting the winner rates layer-wise, an appropriate winner rate is chosen for each layer with the accuracy drop less than a certain threshold. In short, winner rate searching is negligible compared to training.
>
> Here, we show the experiment results on ResNet-32 layer-wise. The chosen winner rates for the first 31 layers except the last fully connected layer are:
> [0.5, 0.3, 0.3, 0.4, 0.3, 0.4, 0.3, 0.1, 0.4, 0.1, 0.3, 0.3, 0.3, 0.3, 0.1, 0.5, 0.3, 0.4, 0.4, 0.3, 0.3, 0.5, 0.4, 0.5, 0.5, 0.5, 0.3, 0.2, 0.1, 0.2, 0.1].
> These activation winner rates are applied on the same weight-pruned ResNet-32 as in the paper, we can get an improved IPnet for ResNet-32 with a 94.61% accuracy on CIFAR-10 with 11.6% left MACs as in the following table. Better accuracy and better computation reduction are both obtained.
>
> Approach 	MAC %	Accuracy drop
> Group-wise	13.7%	-0.43%
> Layer-wise	11.6%	-0.40%
>
> -- Comparison
>
> Thanks to provide related references. We’ll include them in related works. After thoroughly reading these 3 papers, the comparison table is shown as follows. In “Weight %” and “MAC %”, the less the better. All comparisons are conducted based on similar model structures for the same dataset. As seen from the table, integral pruning achieves the best computation reduction with marginal effects on model accuracies.
> While the existing feature map pruning is friendly to conventional hardware platforms, our IP method needs specific accelerator designs to fully utilize the significantly reduced computation cost. We hope the IP method can inspire DNN accelerator designs, and indeed our hardware project is ongoing to fulfill the potential from the proposed IP algorithm.
>
> Dataset	         Model	               Weight %	MAC %	  Accuracy drop
> *****************************************************
> MNIST	         MLP-3 (ours)       10%	                3.65%	  +0.01%
> 	                 MLP-4 [1]	       15.6%	        -	          -0.06%
> *****************************************************
> ImageNet	AlexNet (ours)	38.8%	        28.9%	  +0.04%
> 	                VGG-A [1]	        17.5%	        69.6%	  +0.03%
> 	                VGG-16 [2]	        94.4%	        24.8%	  -3.93%
> *****************************************************
> CIFAR-10	ResNet-32 (ours)	32.4%	        13.7%	  -0.43%
> 	                ResNet-164 [2]	84.8%	         52%	  -0.5%
> 		        ResNet-164 [2]      48.5%	         36%	  -1.0%
> 	                ResNet-20 [3]	62.8%	         -	          -1.1%
> *****************************************************
> [1] Zhuang Liu, Jianguo Li, Zhiqiang Shen, Gao Huang, Shoumeng Yan, and Changshui Zhang. Learning efficient convolutional networks through network slimming. ICCV 2017.
> [2] Zehao Huang and Naiyan Wang. Data-driven sparse structure selection for deep neural networks. ECCV 2018.
> [3] Jianbo Ye, Xin Lu, Zhe Lin, and James Z Wang. Rethinking the smaller-norm-less-informative assumption in channel pruning of convolution layers. ICLR 2018.
>
> -- There is no precise statement somewhere.
>
> 1) The effects on accuracy are summarized in Table 1 at the beginning of Section 4. We’ll also include clear statements in subsections.
> 2) Fig.4 and Fig.5 are targeted for two discussion issues. Fig.4 is to show the advantage of the proposed dynamic activation pruning compared to the static solution. Fig.5 is to give an example for winner rates selection. We will have subsections to make them clearly separable.
> 3) We will double check to avoid any unclear statements and typos.

---

> > ### Comment · AnonReviewer3 · 2018-11-26
> > **Reply to author's response**
> >
> > The authors have commented on the major issues regarding the time complexity on selection of winner rate per layer, compared their method against existing channel/layer based pruning methods and agreed to correct few minor issues. The authors have empirically observed that searching for the right set of choices for winner rate at every layer are computationally efficient and considered negligible when compared to entire training time.
> >
> > Although the authors compare their method against existing pruning techniques in terms of pruned weights and saved MAC operations, the numbers reported from their method ar only valid on dedicated DNN accelerator design platform,s but not on conventional hardware. I would like to clearly state that the comparison of MAC's against existing techniques would make sense only if it is computed based upon the conventional hardware settings. As stated in the conclusion of the article and based on the above set of comparisons, a specially designed hardware is essential to leverage the activation sparsity induced by their method. I would highly recommend the authors to compare the inference time of all the networks (including unpruned network, pruned networks from existing pruning techniques and networks obtained from their method) on their specially designed dedicated DNN accelerator hardware platforms. Besides, I would suggest the authors to compare similar architectures under different techniques, for e.g. VGG baseline, VGG on existing techniques and VGG with their IP technique. These comparisons would support claim of the paper. Finally, the take-away message from the current version of the paper is not very clear from the numbers or comparisons and might not be interesting for the audience of ICLR. I would not revise my rating and reject this submission.

---

### Official Review · AnonReviewer2 · 2018-11-03
**A simple network compression strategy combining weight and activation pruning.**

**Rating:** 5
**Confidence:** 4

**Review:**

The main contribution of the paper is an integral model compression method that handles both weight and activation pruning. Increasing the network weight and activation sparsity can lead to more efficient network computation.  The authors show in the paper that pruning the network weights alone may result in a decrease in activation sparsity, which may not necessarily improve the overall computation. The proposed solution is a 2-stage process that first prunes the weights and then the activation.

Pros:

- The results show that the proposed method is effective in reducing the number of multiply-and-accumulate (MAC) compared to weight pruning alone. The improvements are consistent across multiple network architectures and datasets.
- It also shows that weight pruning alone leads to a slight increase in the number of non-zeros activation.

Cons:

- A simple approach with limited novelty.
- Related work should include other compression techniques, such as low-rank approximation,  weight quantization and varying hidden layer sizes.
- There is no comparison with other model compression techniques mentioned above.

---

> ### Author Response · Authors · 2018-11-12
> **Novelty and Comparison with Related Works**
>
> Thanks for your reviews.
>
> -- Novelty
>
> Our two key contributions are 1) to explore the sparsity limit in both weight and activation and 2) the idea of dynamic activation masks to prune unimportant information in neuron responses.
>
> Firstly, the integration of static weight masks and dynamic activation masks reduces the computation cost significantly, which will give a great potential to specified accelerator designs as claimed in our conclusion. The second key contribution on activation pruning is our major novelty. The activation masks are easy to implement and greatly save computation cost. Furthermore, our proposed activation pruning method remedies the activation sparsity loss for weight pruned models, and it’s also feasible on non-ReLU functions.
>
> -- Comparison with Related Works
>
> Thanks for your suggestion. We shall include the discussion about other compression techniques in our related works.
> We have two concerns here:
> 1) Our proposed activation pruning method is orthogonal to many compression techniques, such weight matrix decomposition, weight quantization. The reason why we focus on weight pruning is that we are aiming to explore the sparsity limit in DNNs.
> 2) On the other hand, our activation pruning approach can be aligned with the topic about feature map pruning. We add the comparison with some typical papers here. In “Weight %” and “MAC %”, the less the better. Our IPnets achieve the largest MAC reduction while model accuracy is basically not compromised.
>
> Dataset	         Model	               Weight %	MAC %	  Accuracy drop
> *****************************************************
> MNIST	         MLP-3 (ours)       10%	                3.65%	  +0.01%
> 	                 MLP-4 [1]	       15.6%	        -	          -0.06%
> *****************************************************
> ImageNet	AlexNet (ours)	38.8%	        28.9%	  +0.04%
> 	                VGG-A [1]	        17.5%	        69.6%	  +0.03%
> 	                VGG-16 [2]	        94.4%	        24.8%	  -3.93%
> *****************************************************
> CIFAR-10	ResNet-32 (ours)	32.4%	        13.7%	  -0.43%
> 	                ResNet-164 [2]	84.8%	         52%	  -0.5%
> 		        ResNet-164 [2]      48.5%	         36%	  -1.0%
> 	                ResNet-20 [3]	62.8%	         -	          -1.1%
> *****************************************************
> [1] Zhuang Liu, Jianguo Li, Zhiqiang Shen, Gao Huang, Shoumeng Yan, and Changshui Zhang. Learning efficient convolutional networks through network slimming. ICCV 2017.
> [2] Zehao Huang and Naiyan Wang. Data-driven sparse structure selection for deep neural networks. ECCV 2018.
> [3] Jianbo Ye, Xin Lu, Zhe Lin, and James Z Wang. Rethinking the smaller-norm-less-informative assumption in channel pruning of convolution layers. ICLR 2018.

---

### Official Review · AnonReviewer1 · 2018-11-03
**Simple idea and lack of experiments**

**Rating:** 4
**Confidence:** 3

**Review:**

This paper proposes to compress the deep learning model using both activation pruning and weight pruning. Combining both sparsities, the MACs are significantly reduced.

My main concern is that there is no time comparison. The experiments only show the reduction in terms of the number of non-zeros in weights and activation as well as the MACs. Typically, to deal with sparse activations and sparse weights, there are some overhead computations such as computing indices. Also, dense matrix-matrix(vector) multiplications can be faster by using specially designed libraries.  I would suggest the authors show the improvement for the proposed compression approach in terms of wall-clock time, in CPU, GPU or other hardware platforms.

The pruning method seems straight-forward to me. I am wondering how to choose the winner rate for each layer. It seems to take a quite long time to pick a set of winner rates for a deep neural network.

The paper is easy to read in general. However, it is not clear to me how such a compression approach can speed up the training or the inference of deep learning models in practice.

---

> ### Author Response · Authors · 2018-11-12
> **Speedup Test and Winner Rates Searching Time**
>
> Thanks for your reviews. We have some supplementary experiment results here, and hope these can address your concerns.
>
> -- Time comparison
>
> 1) As claimed in the conclusion of this paper, the proposed integral pruning approach is targeted for application specific integrated circuit (ASIC) designs with efficient sparse matrix computation supports. Like the approach in [1][2] where the deep compression method inspires a specific accelerator design, the significant save of computation cost in our IPnets indicates the great potential of efficient ASIC designs in terms of energy and speed.
> [1] Han, S., Mao, H. and Dally, W.J., 2015. Deep compression: Compressing deep neural networks with pruning, trained quantization and huffman coding. arXiv preprint arXiv:1510.00149.
> [2] Han, S., Liu, X., Mao, H., Pu, J., Pedram, A., Horowitz, M.A. and Dally, W.J., 2016, June. EIE: efficient inference engine on compressed deep neural network. In Computer Architecture (ISCA), 2016 ACM/IEEE 43rd Annual International Symposium on (pp. 243-254). IEEE.
>
> 2) While we are working on the accelerator design to fully exploit integral pruning, the speedup on fully-connected (fc) layers benefiting from activation pruning is easy to be demonstrated on conventional computation platforms, such as a desktop CPU. This is because after activation pruning, the weight matrix of fc layers can be structured condensed by removing all connections related to the pruned activations. We take the last 3 fc layers of AlexNet on Imagenet dataset, and the experiment setup is shown in Table I. Batch size is 1 here, which is the case we care about most in real-time applications on edge devices.
>
> Table I. Experiment setup:
> Framework	          CPU	                   Memory	Batch size
> TensorFlow 1.10	  Intel i7-7700HQ	   8 GB	        1
>
> The input activations can be pruned without compromising accuracy as shown in table II. Note that in Table II, the time per layer with activation pruning mainly comprises I) argpartition on input activation vector and II) matrix computation on the condensed layer. A 1.95x ~ 3.65x speedup is achieved. Time spent on argpartition to get winner activations is also included, which accounts for a small portion compared to the time spent on previous dense layers.
>
> Table II. Measurement results:
> Layer	Size	               Input acti %	                  Time per layer	                                     argpartition  (msec)	     Speedup
> 		                                                    Dense (msec)	With acti pruning (msec)
> Fc1         	9216x4096     27.70%	            10.19975901	3.95335722	                             0.879592419	                     2.58 X
> Fc2         	4096x4096	10%	                    4.544641018	1.24421525	                             0.524742842	                     3.65 X
> Fc3         	4096x1000	10%	                    1.520430803	0.778268337	                             0.397073746	                     1.95 X
>
> -- Winner Rates Searching Time
>
> 1) As discussed in Section 3.2, the winner rate per layer is empirically chosen that the accuracy drop is less than a certain threshold on a validation set. This criterion has been thoroughly verified by the experiments on various datasets and models as in Section 4.
> 2) The time spent on selecting winner rates can be negligible compared to training time.
> By wall-clock time measurements, scanning time over winner rates by using TITAN Xp with 12G memory is:
> For Fig.5 (a), 1 h 12 min; for Fig.5 (b), 20 min.
> For super deep NN structure such as ResNet-152, like distributed training, the winner rate scanning can be accelerated by GPUs working in parallel. On the other hand, the time spent on winner rate searching doesn’t hinder the inference time.

---

### Meta-Review · Area_Chair1 · 2018-12-17
**lack of novelty**

**Confidence:** 5
**Recommendation:** Reject

**Metareview:**

This paper proposes to compress the deep learning model using both activation pruning and weight pruning. The reviewers have a consensus on rejection due to lack of novelty.